# Confidence does not mediate a relationship between owner experience and likelihood of using weight management approaches for native ponies

Ashley B. Ward[1,2]*, Patricia A. Harris[3], Caroline McG. Argo[1], Christine A. Watson[1], Neil M. Burns[4], Madalina Neacsu[2], Wendy R. Russell[2], Dai Grove-White[5], Philippa K. Morrison[1]

1 Scotland's Rural College, Bucksburn, Aberdeen, United Kingdom, 2 School of Medicine, Medical Sciences and Nutrition, The Rowett Institute, University of Aberdeen, Foresterhill, Aberdeen, United Kingdom, 3 Equine Studies Group, Waltham Petcare Science Institute, Leicestershire, United Kingdom, 4 Department of Rural Economy, Environment and Society, Scotland's Rural College, Edinburgh, United Kingdom, 5 Faculty of Health and Life Sciences, University of Liverpool, Wirral, United Kingdom

* ashley.ward@sruc.ac.uk

**Data Availability Statement:** All data files are available from SRUC's FigShare repository (DOI:10.58073/SRUC.22608412).

## Abstract

Native ponies are at increased risk of obesity and metabolic perturbations, such as insulin dysregulation (ID), a key risk factor for endocrinopathic laminitis. Management and feeding practices can be adapted to maintain healthy body condition and support metabolic health, but owners may inadvertently provide their ponies with inappropriate management leading to obesity and exacerbating risk of metabolic disease. Adoption of preventative weight management approaches (WMAs), including regular monitoring of body condition, providing appropriate preserved forage, promoting seasonal weight loss, and using exercise accordingly, are key in supporting native ponies' metabolic health. The factors influencing the adoption of WMAs, such as owners' experience and confidence, require exploration. The aim of the current study was to understand factors influencing owners' likelihood to undertake certain WMAs, to develop our understanding of suitable intervention targets. A total of 571 responses to an online cross-sectional questionnaire were analysed. Mediation analysis revealed that whilst long term (≥20 years) experience caring for native ponies was associated with owners increased, self-reported confidence in identifying disease and managing their native ponies, this did not translate to an increased likelihood of implementing WMAs. Conversely, respondents who managed ponies with dietary requirements related to obesity, laminitis, or equine metabolic syndrome were more likely to use WMAs related to feeding, seasonal weight management and exercise. Owner confidence was assessed and rejected as a mediator of the relationship between experience and WMA use. These results highlight the need for further work that elucidates the pathways leading owners to undertake action against obesity without the need for ponies to develop overt disease, as well as suggesting a need for long term managers of native ponies to update management practices with preventative care as the focus.

**Funding:** This study was funded by Mars Petcare and is part of a PhD studentship funded by the Scottish Funding Council Research Excellence Grant (REG). Authors WR and MN receive salary support from the Rural and Environment Science and Analytical Services Division (RESAS). With the exception of PH (employed by the funding organization), the funding organization did not have any additional role in the conceptualization, methodology, investigation, data curation, formal analysis, decision to publish, or preparation of the manuscript. PH was involved in study design, data interpretation, and manuscript preparation.

**Competing interests:** Co-author PH is employed by the funding organization. This does not alter our adherence to PLOS ONE policies on sharing data and materials.

## 1. Introduction

Obesity continues to dominate discussions around current equine welfare issues [1, 2], particularly among the UK's leisure horse and pony population, where the prevalence of obesity ranges from 21% to 35% [3–5]. Native breed-type ponies face a heightened risk of obesity [5, 6], likely due to the combination of genetic, environmental and phenotypic factors influencing energy metabolism [7–9]. Such factors are thought to be responsible for these breeds' increased risk of insulin dysregulation (ID) [10–12], a metabolic perturbation often associated with pituitary pars intermedia dysfunction (PPID) [13, 14], and central to the diagnosis of equine metabolic syndrome (EMS) [15]. Insulin dysregulation significantly increases ponies' risk of endocrinopathic laminitis [6, 16], one of the most pressing welfare concerns associated with ID. However, obesity can have direct detrimental effects on cardiovascular health and fertility [17, 18], and importantly is a risk factor for ID [11]. Obesity also impairs recovery from laminitis [19]. Addressing obesity in British native ponies is imperative to enhance the welfare of a large proportion of equids in UK, and globally [20].

Weight management approaches (WMAs) that help induce weight loss and improve insulin sensitivity in the horse have been extensively researched. Types of WMAs include dietary restriction and provision of forages that are low in Non-Structural Carbohydrates (NSCs), which can induce body mass losses in ponies under controlled study conditions [21–24]. Of particular importance for native ponies, whose metabolism adaptively slows during colder seasons to conserve energy, are dietary restriction and low NSC carbohydrate feeding which can be used as a WMA to replicate the natural reductions in available nutrients during cold seasons [25]. Incorporating exercise into diet-restricted regimens may also have beneficial effects upon insulin sensitivity [26], thus reducing the likelihood of hyperinsulinaemic spikes which may exacerbate ID, and increase the risk of laminitis.

Positive results of equine weight-loss interventions have been demonstrated in the field [27], and scientific findings have been translated into freely available guides that are offered by various charities, such as the Blue Cross [28], The Horse Trust [29], and World Horse Welfare [30]. These weight management guides describe WMAs in categories relating to monitoring body condition, providing appropriate feeding, promoting seasonal weight loss, providing the appropriate exercise, and controlling access to pasture. Widespread use of proposed WMAs could help to reduce obesity across the leisure horse population. However, the success of equine weight loss intervention is largely determined by owners' compliance [27], along with the horses' access to pasture.

Owner compliance with weight-loss regimens and adherence to recommended WMAs may be influenced by a combination of behavioural, social, and physical factors [31]. Recreational horse owner beliefs' on the appropriate management for their animals appears to be linked to knowledge and experience [32], both of which are modifiable factors, suggesting a role for targeted education and training to promote the best practices for native pony weight management. Effective behavioural intervention through education requires that the target group identify their own requirement for such training. However, research regarding horse owners' identification of obesity in their own horses has shown that many owners may have difficulty doing this [33, 34]. Attitudes driven by subjective knowledge gained in the equestrian field, such as holding differing thresholds for acceptable body condition based on equestrian discipline [35], and the mistrust of practices involving dietary restriction [34], may involve learned behaviours nested in tradition which no longer serve the modern management of native ponies.

Confidence in abilities drawn from subjective experience could make owners less likely to seek or accept changing approaches to management. At its most extreme, evidence of this

behavioural phenomenon can be seen in relation to the COVID-19 pandemic, where subjective and biased information sourcing appeared to be linked with scepticism towards governmental health recommendations [36]. In the present context, non-evidence-based knowledge acquisition leading to inappropriate native pony management could be linked with a worsening obesity problem for UK leisure equids. Understanding how owner experience influences confidence, and how this confidence in turn is associated with the implementation of WMAs, could reveal insights into the drivers of owner behaviour in relation to promoting a healthy weight in their animals.

A relatively novel approach to understanding interconnected behavioural themes in the animal and veterinary sciences is the application of structural equation modelling (SEM) to identify latent concepts underlying behavioural data gathered from questionnaires. Latent concepts are effects present in the data which cannot be directly measured. This can extend from physical attributes such as gait and morphology [37], to behavioural considerations such, as empathy and client needs [38]. Under the umbrella of SEM methodologies, mediation analysis can be applied to evaluate relationships between measured or latent variables, where these relationships are hypothesised to have both direct and indirect pathways of association [39]. Such analysis has previously been applied to evaluate the effect of client satisfaction upon the relationship between perceived pain and the decision to euthanise dogs with osteoarthritis [40].

The aim of the present study was to explore the role played by owner experience and confidence in integrating evidence-based WMAs into the management of their native breed-type ponies. This was addressed by the development of a causal model of owners' experience and their undertaking of WMAs in four areas relating to weight management, including: monitoring body condition, providing appropriate feeding, promoting seasonal weight loss, and promoting exercise. Subsequently, mediation analysis was employed to evaluate the direct effect of native pony owners experience upon their likelihood of using WMAs, and the indirect effect of experience upon WMA usage via confidence.

## 2. Methods

### 2.1. Questionnaire design

A cross-sectional survey was designed to collect information on all aspects of the general management of native ponies in Scotland. Ethical approval for the study was obtained from SRUCs Social Science ethics committee on the 30th of January 2020. The survey was made available for data collection immediately after ethical approval was obtained until the 13th of October 2021. Informed consent was confirmed through respondent completion of a tick box question at the beginning of the survey. The questionnaire included 77 questions in the areas of personal experience, general management, weight management, feeding and nutrition, exercise and laminitis care. The entire survey included a variety of binary, Likert scale items, multiple response, ranking and open text formats. The present analysis focused on binary variables relating to experience and WMAs and 5-point Likert scales related to confidence. Survey questions (excluding those related to pasture management which were outside the scope of the present study) can be found in the S2 File.

### 2.2. Data collection

The anonymised online survey was created using Smart Survey software and was promoted by press release and social media activity over an 18-month period. Respondents were entered into a prize draw to win a 100 GBP gift voucher as an incentive to undertake the survey. Upon survey closure, one respondent was selected at random for receipt of this incentive prize, after

which point identifiable data were permanently deleted. Respondents identified as residing outside of Scotland or those that did not own and care for native ponies were excluded from further analysis based on answers to two screening questions.

## 2.3. Data preparation

One of the key outcomes measured was the number of years owners had been involved in native pony management, with the aim to compare responses from owners with more experience of native pony management compared to those with less experience. In order to do this sensitivity analysis was performed with single and multiple explanatory variables specifying experience alone, and confidence + experience as predictors of the independent WMA variables. The analysis was repeated with the "Years of Experience" variable dichotomized into equal to or greater than four, ten, twenty and thirty years of experience (for category distribution see Table 1). The 20-year cut off for dichotomisation yielded a greater number of statistically significant relationships between experience and the outcomes tested than other groupings (S1 Table in S1 File). To explore these relationships further, years of experience was converted to a binary variable by defining owners with less than or greater than 20 years of experience ($EXP_{YEARS}$). It was considered that for the purposes of this study, these groups adequately represented owners with extensive experience caring for native ponies (>20 years), and those within a range of experience up to 20 years. The resulting binary categories preserved the higher proportion of owners at the upper extreme whilst providing a suitable proportion of owners with fewer years of experience to allow for fair comparison.

Owners were also categorised based on their response to a question asking which clinical conditions or dietary needs influenced their feeding practices. A new binary variable was created to categorize owners who self-reported managing their ponies' nutrition for metabolic disease-related conditions, compared to those who did not ($EXP_{MET}$). Dichotomisation was achieved by assigning "Yes" to owners who indicated any of laminitis, equine metabolic syndrome (EMS) or weight management as conditions that influenced their horses' feeding, and "No" to those who did not select these options. Questions relating to WMAs were presented to respondents in a nominal, single choice format. These variables were then converted to binary variables to allow comparison between those who undertook a given weight management approach and those who did not. The original nominal variable categories are summarised along with the final binary versions in the results.

Owners were asked about the frequency of, and methods used for, the assessment of body weight and body condition. In response to the question on weight, where owners selected methods involving a calibrated weighbridge, or morphometric measurement of some part of the horses' body with calculation, they were assigned as using a "standardised" method for body weight assessment. Visual assessment and methods based on body condition assessment were categorised as non-standardised methods. In relation to body condition, standardised assessment of body condition was assigned where owners selected a method of assessment involving the use of either a six-point (0–5) [41] or a nine-point (1–9) scale [42], or morphometric measurements combined with the use of an online body condition calculator tool. Visual assessment and feeling the body were classified as non-standardised assessment of body condition.

Eight Likert Scale items each on a 5 –point scale were used to measure owners' self-reported confidence in various aspects of their ponies' management, and six items measured owners' self-reported confidence in identifying the signs of health conditions affecting the horse. For mediation analysis, the Likert items were summed to create two latent continuous Likert Scale variables. The first latent variable represented "Confidence in management" (CONF

**Table 1. Descriptive statistics of respondent demographics and equestrian experience.**

| Variable | Proportion of total (n = 571) (n (% [95% CI])) |
|---|---|
| Years of caring for native ponies | |
| 0–3 | 41 (7% [2.972, 11.534]) |
| 4–9 | 103 (18% [13.811, 22.373]) |
| 10–19 | 161(28% [23.951, 35.513]) |
| 20–29 | 117 (20% [16.259, 24.821]) |
| >30 | 150 (26% [22.080, 30.590]) |
| **All ≥ 20 years (EXP $_{YEARS}$)** | **267 (47%)** |
| *Feeding ponies for specific health condition | |
| Laminitis | 218 (38% [35.00, 43.626]) |
| Weight management | 145 (25% [21.903, 30.520]) |
| Equine metabolic syndrome | 86 (15% [11.311, 19.928]) |
| Equine gastric ulcer syndrome | 36 (6% [0.023, 0.110]) |
| Respiratory condition | 36 (6% [0.023, 0.110]) |
| Colic | 36 (6% [0.023, 0.110]) |
| **Managing any of EMS/laminitis/weight (EXP $_{MET}$)** | **305 (53%)** |
| Frequency of body condition assessment | |
| Daily | 160 (28% [23.601, 32.367]) |
| Weekly | 241 (42% [37.762, 46.528]) |
| Monthly | 126 (22% [17.657, 26.423]) |
| Annually | 3 (1% [0, 4.919]) |
| Occasionally | 42 (7% [2.972, 11.738]) |
| Never | 0 |
| **More frequent than annual assessment (WMA $_{BCS\ FREQ}$)** | **526 (92%)** |
| *Method used for body condition assessment | |
| Visual assessment | 298 (52% [48.077, 56.430]) |
| Morphometric measurement | 118 (21% [16.608, 24.961]) |
| BCS (1–5 scale) | 90 (16% [11.714, 20.066]) |
| BCS (1–9 scale) | 42 (7% [3.322, 11.674]) |
| Other | 24 (4% [0.175, 8.528]) |
| **Standardised method (WMA $_{BCS\ METHOD}$)** | **250 (44%)** |
| Analysed preserved forage | |
| **Yes (WMA $_{PRES\ FORAGE\ TEST}$)** | **74 (13% [9.266, 16.910])** |
| No | 338 (68% [64.161, 71.805]) |
| Unknown | 110 (19% [15.559, 23.204]) |
| Preserved forage | |
| Hay | 275 (48% [44.056, 52.402]) |
| Mixture of hay and haylage | 98 (17% [13.112, 21.458]) |
| **Soaked hay (WMA $_{PRES\ FORAGE\ SOAK}$)** | **94 (16% [12.413, 20.759])** |
| Haylage | 68 (12% [7.867, 16.213]) |
| Steamed hay | 22 (4% [<0.001, 8.171]) |
| Short chop fibre source | 9 (2% [<0.001, 5.899]) |
| None | 6 (1% [0.001, 5.374]) |
| Weight management approach (Autumn) | |
| Maintain | 371 (64% [61.014, 68.949]) |
| **Encourage loss (WMA $_{SEASON\ AUT}$)** | **153 (27% [22.902, 30.837])** |
| Encourage gain | 48 (8% [4.545, 12.481]) |
| Weight management approach (Winter) | |

*(Continued)*

**Table 1.** (Continued)

| Variable | Proportion of total (n = 571) (n (% [95% CI])) |
|---|---|
| Maintain | 330 (58% [523.671, 62.057]) |
| **Encourage loss (WMA $_{SEASON\ WINT}$)** | **201 (35% [31.119, 39.504])** |
| Encourage gain | 41 (7% [3.147, 11.532]) |
| Most important aspect of exercising pony | |
| Horses' performance | 172 (30% [25.350, 33.933]) |
| Personal enjoyment | 169 (30% [25.874, 34.458]) |
| **Weight management (WMA $_{EXERCISE}$)** | 104 (18% [13.986, 22.570]) |
| Boredom relief (horses') | 88 (15% [11.189, 19.773]) |
| Other | 28 (5% [0.699, 9.283]). |
| Not applicable to my horses | 11 (2% [0.000, 6.311]) |

\* Multiple response question (the sum of proportions >1.0)

$^{i}$ Variables in bold are included in mediation analysis

$_{MANAGEMENT}$); the sum of 8 Likert items relating to management of native ponies. The second latent variable represented "Confidence in identifying signs of disease" (CONF $_{DISEASE}$); the sum of 6 Likert items relating to owners' confidence in identifying the signs of obesity, laminitis, EMS, PPID, colic and loss of body condition. Composite scores of the latent Likert Scales were assumed to represent the underlying continuous distribution of confidence.

Initial data screening was performed in Excel version 2301. Data re-coding, formatting, visualisation, and analysis were conducted using R version 4.2 [43]. The full code used for analysis can be viewed in the supporting information (S1 File).

## 2.4. Data analysis

Descriptive statistics were performed with proportions and 95% confidence interval approximation to summarise categorical variables [44]. Cronbach's Alpha statistic was calculated as a measure of similarity within the scale items and was used to evaluate whether the items of scales relating to the two latent variables, namely CONF $_{MANAGEMENT}$ and CONF $_{DISEASE}$ reflected their respective constructs. Skewedness and kurtosis of these continuous latent variables was assessed. Both variables demonstrated right skew, and so to minimise the impact of this violation of the normality assumption, statistical inference was made from confidence intervals derived from 5000 bootstrap samples [45].

Mediation analysis was performed using the R package "mediation" [46]. Models were constructed to simultaneously assess the direct effect of the independent variable (IV) "EXP" on the dependent variable (DV) likelihood of WMAs, and the indirect effect of the IV on the DV through the latent mediator "Conf" [39]. Outcome models were fitted using binary probit generalised linear models (GLMs), and the mediator models were fitted via linear regression of the effect of experience on the latent scales of confidence. Bias-corrected bootstrap-based confidence intervals were used to examine the indirect effects of experience on likelihood of undertaking WMAs via confidence. Fig 1 shows a directional acyclic graph outlining the shared structure of the models outlined.

Relationships between binary and categorical variables assessed outside of mediation analysis were modelled via GLM using a logit link function, or Chi Square test of independence. Results for regressions are reported with beta coefficients (b), 95% confidence intervals (95% CIs) and p-values (p).

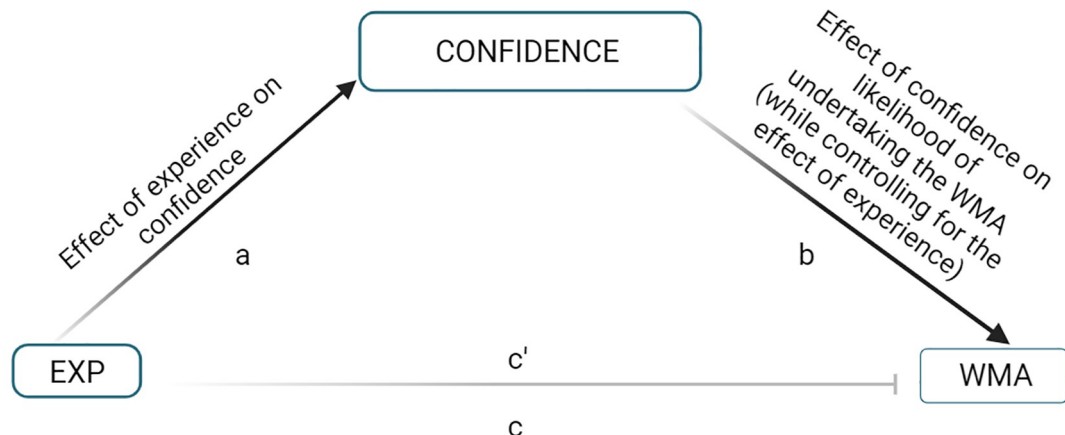

c'= Effect of experience on likelihood of undertaking the WMA (while controlling for confidence)
c= Effect of experience on likelihood of undertaking the WMA

Indirect effect / Average causal mediated effect = c-c' (Tingley et al, 2014)
Average direct effect = c'

**Fig 1. Directional acyclic graph demonstrating mediation model structures.** The effects of mediation on the relationships between explanatory and outcome variables were evaluated simultaneously through estimating combined associations between experience (EXP) and mediator, and between mediator and weight management approach (WMA). Mediation was assessed through evaluating the indirect effect (pathway c') [39] and comparing the strength and direction of this effect relative to the average direct effect (pathway c). Dark lines with arrow heads indicate significant relationships, whilst grey lines with blunt ends indicate non-significant relationships. Figure created with Biorender.com.

## 3. Results

The survey received a total of 1779 responses. All rows (responses) with missing data for questions included in the analysis were excluded, leaving a final sample size of n = 571. The sample was comprised largely of leisure owners (71%) as opposed to industry professionals (defined from questionnaire *), and the majority of the sample respondents did not hold industry related qualifications (70%) (Fig 2).

Roughly half (53%) of owners managed the dietary requirements of a pony that was overweight and/or had laminitis, and/or EMS, and almost half (47%) had over 20 years of experience managing native ponies. A full summary of the sample can be seen in S2 Table (S1 File). See Table 1 for summary statistics pertinent to the proceeding analysis.

In order of most frequent to least, owners reported being "Very" confident in; pasture management (61%), rugging/ clipping/ stabling (60%), controlling access to pasture (56%), herd management (54%), providing the appropriate type of preserved forage (50%), providing the appropriate type of complementary feeding (50%), providing the appropriate amount of supplementary feeding (48%), and providing the appropriate amount of preserved forage (48%) (Fig 3). Owner confidence in identifying signs of disease was more variable than confidence in management practices. The highest confidence rating was most often indicated for identifying; obesity (72%), loss of condition (65%), laminitis (59%), colic (58%), PPID (26%), and EMS (20%). Respondents expressed markedly lower confidence in identifying the signs of EMS and PPID in comparison to identifying the signs of other conditions, and any management practices (Fig 3).

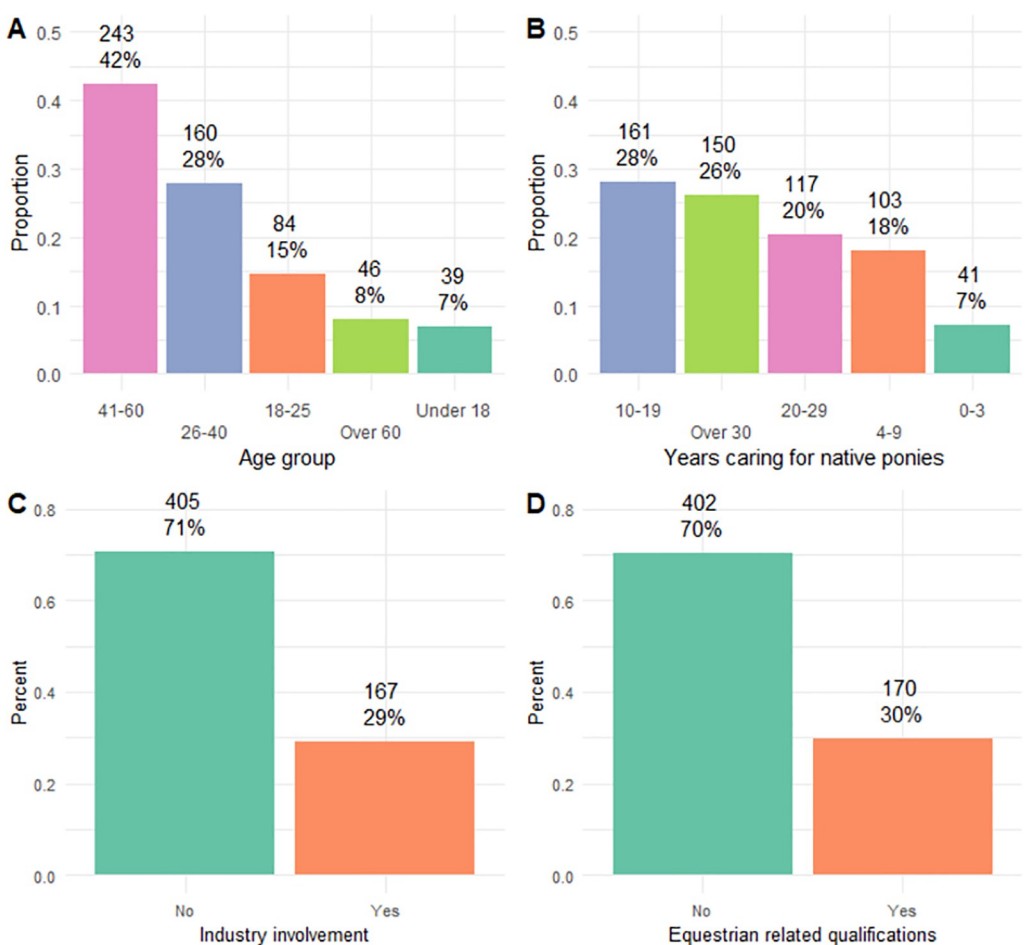

**Fig 2. Respondent demographic information.** Distribution of native pony owners by (A) age, (B) years of experience managing native ponies', (C) industry involvement, and (D) industry related qualifications.

Latent confidence variables formed from the sum of categorical responses demonstrated left skew toward the higher confidence (Table 2). This skew was present in both measures of confidence but more so for confidence in management (CONF $_{MANAGEMENT}$), which had a median score more than double that of confidence in identifying signs of disease (CONF $_{DISEASE}$). Cronbach's alpha for CONF $_{MANAGEMENT}$ and CONF $_{DISEASE}$ were 0.948, and 0.745 respectively, indicating "excellent", and "acceptable" levels of similarity amongst scale items (based on an arbitrary scale of Cronbach's alpha statistical meaningfulness to represent respective underlying concepts [48].

Neither CONF $_{DISEASE}$ nor CONF $_{MANAGEMENT}$ were associated with caring for a pony with metabolic disease related dietary requirements (EXP $_{MET}$), but both were significantly associated with having ≥20 years of experience managing native ponies (EXP $_{YEARS}$) (Table 3). Due to the lack of association between EXP $_{MET}$ and either confidence variable, mediation of the relationship between this measure of experience and the likelihood of undertaking the WMAs was rejected.

### 3.1. WMA 1: Monitoring body condition

Overall, weight was most commonly assessed through tape-measure based methods (49%, n = 277). Within this group, the majority made assessments weekly (39%, 107/277) or monthly

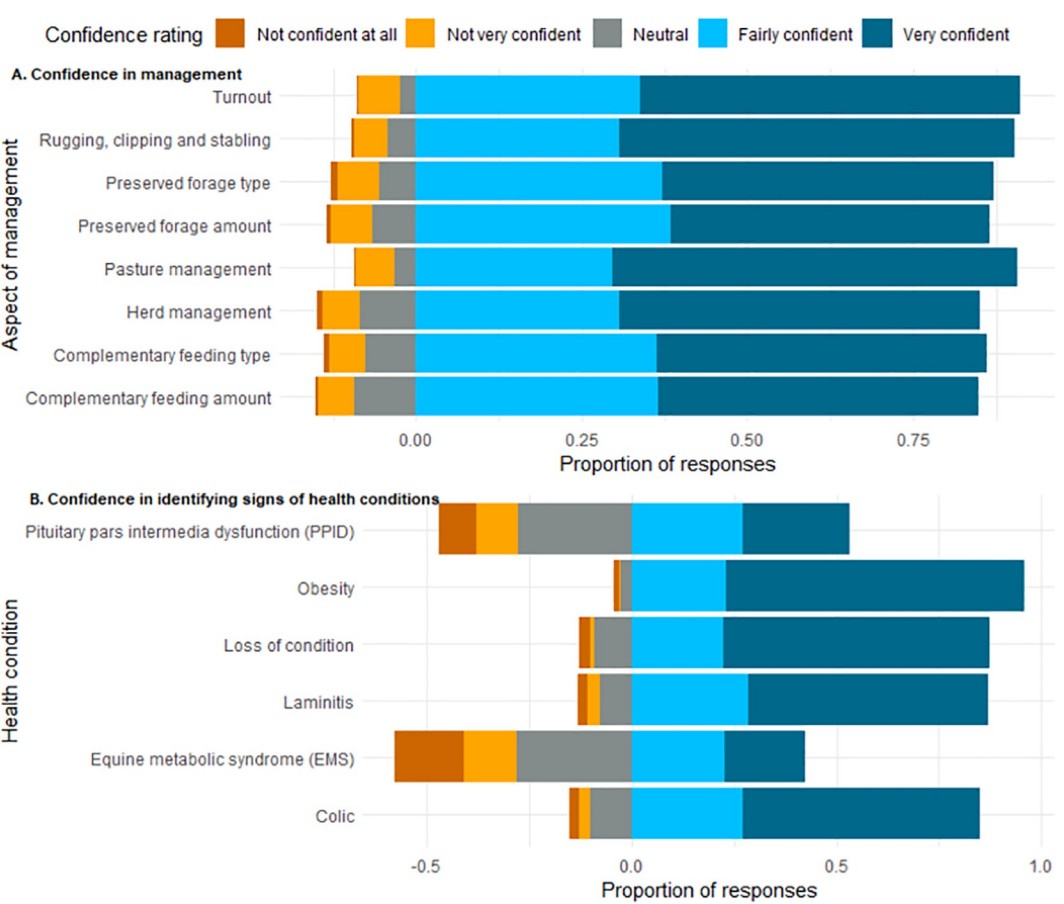

**Fig 3. Percentage of respondents self-reported confidence level in ability to perform management of native ponies, and to identify signs of disease.** Owners ranked Likert style items on a scale of 1–5 with 1 being "not confident at all" and 5 being "very confident". Bars to the left of the central line include 50% of neutral responses and negative scores (e.g. not confident), and to the right of the central line include 50% of neutral responses and positive scores (e.g. fairly or very confident). Plot generated using R package HH [47].

(34%, 93/277). This was followed by visual estimation, which 36% (n = 203/277) used. This approach was most commonly performed either weekly (36%, 73/203) or daily (30%, 60/203). Of those who used a weighbridge to assess weight (9%, n = 50), 38% did so occasionally (19/50), and 30% did so monthly (15/50). Very few used measurements combined with calculation to estimate weight (2%, n = 9). The most common approach to assessing body condition was visual assessment (52%, n = 298), made either weekly (38%, 112/298) or daily (35%, 105/298). Visual assessment of body condition was followed in popularity by morphometric measurement (21%, n = 118), body condition scoring using a 0–5 scale (26%, n = 90), body condition scoring using a 1–9 scale (7%, n = 42), and other methods (4%, n = 24). Where morphometric

**Table 2. Summary of the Likert scales of owners' confidence in managing their native ponies, and confidence in identifying the signs of disease.**

|  | Median; Q1-Q3[a]; Min-max | Skewedness | Kurtosis | Cronbach's Alpha |
|---|---|---|---|---|
| CONF $_{MANAGEMENT}$ (Range: 8–40) | 36; 32–40; 8–40 | -1.540 | 2.393 | 0.948 |
| CONF $_{DISEASE}$ (Range: 6–30) | 17; 15–19; 4–20 | -1.352 | 2.662 | 0.745 |

[a] Q1 – 1ST quartile, Q3- 3rd quartile.

**Table 3. Relationships between confidence scales and the independent experience variables.**

| Outcome | EXP $_{YEARS}$ | | EXP $_{MET}$ | |
|---|---|---|---|---|
| | B (p-value) | 95% CIs | B (p-value) | 95% CIs |
| CONF $_{MANAGEMENT}$ | 1.145 (<0.001) | 0.674, 1.615 | 0.053 (0.870) | -0.407, 0.527 |
| CONF $_{DISEASE}$ | 1.938 (<0.001) | 0.950, 2.881 | -0.570 (0.254) | -1.541, 0.427 |

measurement was used, owners tended to assess weekly (48%, 57/118), or monthly (31%, 36/118). Owners who used BCS systems tended to do so either weekly (0–5 scale- 43%, 39/90; 1–9 scale-57%, 24/42), or daily (0–5 scale- 29%, 26/90; 1–9 scale- 29%, 12/42). These data can be viewed in the Supplementary Materials (S1 Fig in S1 File).

Assessment of body weight was undertaken less frequently by owners adopting a standardised method as opposed to a visual assessment (b = -0.392, 95% CI [-0.130, -0.656], p = 0.003). In contrast, frequency of body condition scoring was not significantly influenced by whether a standardised or visual method was used (b = 0.227, 95% CI [-0.160, 0.621], p = 0.255).

Neither EXP $_{MET}$ or EXP $_{YEARS}$, were significantly directly associated with the likelihood of owners using more frequent (WMA $_{BCS\ FREQ}$) or standardised body condition assessment (WMA $_{BCS\ METHOD}$) (Table 4).

Mediation by CONF $_{DISEASE}$, was assessed for the relationship between EXP $_{YEARS}$ and WMA $_{BCS\ FREQ}$, and the bootstrapped unstandardised indirect effect coefficient was significant (Fig 4A). The total effect of direct and indirect pathways and the average direct effect were negative, but not statistically significant. The indirect effect between EXP $_{YEARS}$ and WMA $_{BCS\ METHOD}$ via CONF $_{DISEASE}$ was statistically significant with a positive coefficient (Fig 4B).

### 3.2. WMA 2: Seasonal weight management

As opposed to aiming for weight loss or gain, the majority of respondents aimed to "maintain" their horses' weight over the autumn (65%), winter (58%), spring (54%), and summer (55%) months (Fig 5). Weight management approach was dependent upon season ($X^2$ (df) = 57.35 (6), $p$ <0.001). Post hoc pairwise comparisons revealed that in the autumn months, owners were significantly more likely to promote weight gain in their ponies than in spring (p<0.001), summer (p<0.001), and winter (p = 0.022).

As previously indicated (Table 3), the latent CONF $_{MANAGEMENT}$ measure was significantly associated with EXP $_{YEARS}$ but not with EXP $_{MET}$, and so mediation by confidence was not tested for the relationship between EXP $_{MET}$ and WMAs relating to seasonal management. Independently of confidence, EXP $_{MET}$ was associated with a significantly greater likelihood of promoting weight loss (opposed to promoting weight gain or maintenance) across autumn (WMA $_{SEASON\ AUT}$) and winter (WMA $_{SEASON\ WINT}$) (Table 5).

While EXP $_{YEARS}$ was associated with CONF $_{MANAGEMENT}$ (Table 2), it was not associated with using the WMAs in seasonal management (Table 4). Mediation analysis (S2 Fig in S1 File) revealed no evidence to suggest that confidence mediated a relationship between EXP $_{YEARS}$ and promoting weight loss in the winter (ADE = 0.005, p = 0.900) or autumn (ADE = 0.042, p = 0.690).

**Table 4. Direct relationships between experience WMAs related to body condition monitoring.**

| Outcome | EXP $_{YEARS}$ | | EXP $_{MET}$ | |
|---|---|---|---|---|
| | B (p-value) | 95% CIs | B (p-value) | 95% CIs |
| WMA $_{BCS\ FREQ}$ | -0.093 (0.543) | -0.395, 0.209 | 0.141 (0.358) | -0.160, 0.444 |
| WMA $_{BCS\ METHOD}$ | -0.169 (0.111) | -0.383, 0.032 | 0.121 (0.277) | -0.087, 0.328 |

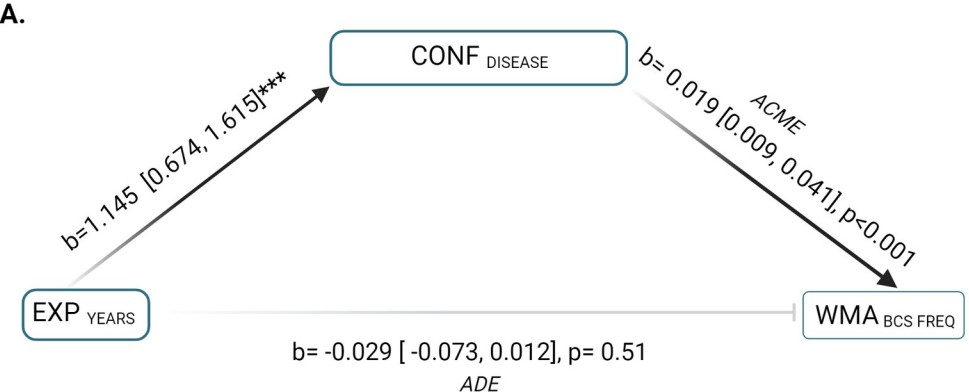

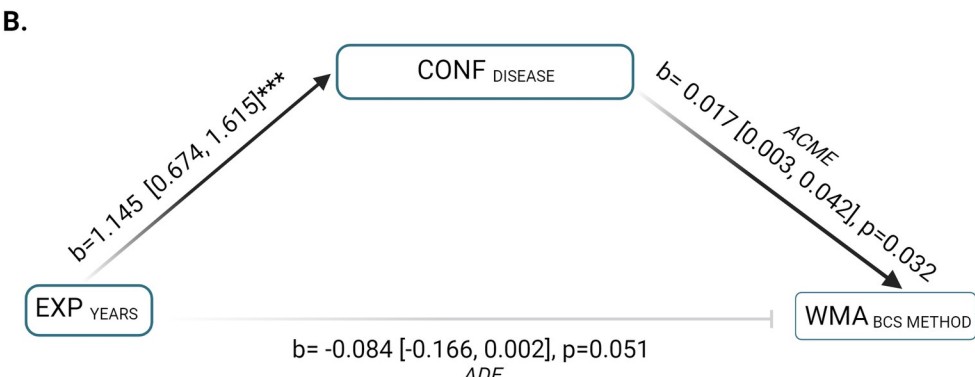

**Fig 4. Mediation model coefficients for the indirect effect (ACME) and average direct effect (ADE) of confidence on the relationship between owners having ≥20 years' experience caring for native ponies (EXP $_{YEARS}$) and WMAs related to monitoring body condition. (A)** EXP $_{YEARS}$ was significantly positively associated with confidence in identifying signs of disease (CONF $_{DISEASE}$). The ACME was significant and positive and ADE was negative and non-significant, demonstrating conflicting effects of the positive indirect pathway between EXP $_{YEARS}$ upon likelihood of more frequent body condition assessment (WMA BCS $_{FREQ}$) via CONF $_{DISEASE}$, and the negative average direct effects of EXP $_{YEARS}$ upon WMA BCS $_{FREQ}$ **(B)** Coefficients between EXP $_{YEARS}$ and owners' likelihood of using standardised methods of body condition assessment WMA BCS $_{METHOD}$ also demonstrated competing direct (negative) and indirect (positive) effects, the ADE was again not significant. Figure created with Biorender.com.

### 3.3. ≥WMA 3: Providing appropriate preserved forage

Between March and August, 56% of owners added preserved forage to their native ponies' diet to supplement their grazing. Provision of preserved forage increased to 95% during the autumn and winter months from September to February. Roughly half of respondents provided their ponies with a vitamin and mineral feed balancer (55%), with the most common reason being to supplement a forage-based diet (38%). Of the 267 owners with ≥20 years of experience, 38 (14%) had undertaken preserved forage analysis compared to 36 of 304 (12%) owners with < 20 years' experience. Probit regression showed that EXP $_{YEARS}$ was significantly associated with a reduced likelihood of undertaking preserved forage analysis, and EXP $_{MET}$ was significantly positively associated with both preserved forage analysis and hay soaking (Table 6).

Mediation analysis showed that the indirect path between EXP $_{YEARS}$ and WMA $_{PRES\ FORAGE\ TEST}$ was not significant (S2 Fig in S1 File), providing no evidence to suggest that CONF$_{MANAGEMENT}$ significantly influenced a relationship between experience and testing preserved forage (ADE = -0.064, p = 0.039). While the average direct effect between experience

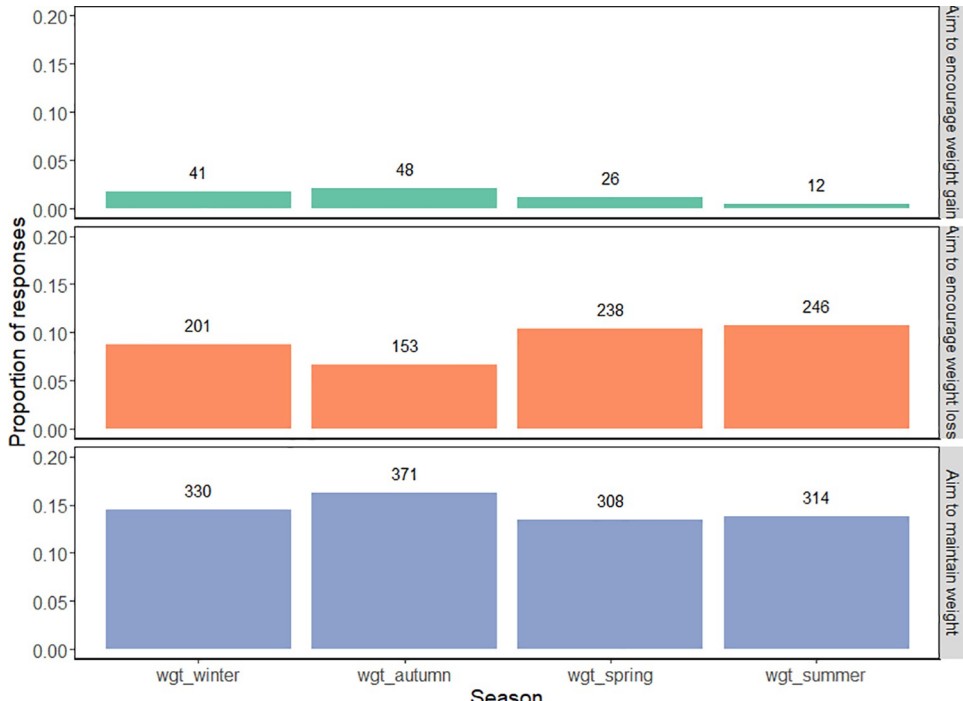

**Fig 5. Weight management approach according to season.** The proportion of native pony owners aiming to promote loss, gain, or maintain their ponies' weight across spring, summer, autumn and winter.

**Table 5. Direct relationships between experience and WMAs relating to seasonal weight management of the horse.**

| Outcome | EXP $_{YEARS}$ | | EXP $_{MET}$ | |
| --- | --- | --- | --- | --- |
| | B (p-value) | 95% CIs | B (p-value) | 95% CIs |
| WMA $_{SEASON WINT}$ | 0.117 (0.255) | -0.095, 0.329 | 0.364 (0.001) | 0.150, 0.579 |
| WMA $_{SEASON AUT}$ | $-0.002e^{-1}$ (0.998) | -0.218, 0.224 | 0.525 (<0.001) | 0.298, 0.754 |

**Table 6. Direct relationships between experience and likelihood of undertaking WMAs related to providing appropriate preserved forage.**

| Outcome | EXP $_{YEARS}$ | | EXP $_{MET}$ | |
| --- | --- | --- | --- | --- |
| | B (p-value) | 95% CIs | B (p-value) | 95% CIs |
| WMA $_{PRES FORAGE TEST}$ | -0.286 (0.024) | -0.538, -0.036 | 0.502 (<0.001) | 0.247, 0.763 |
| WMA $_{PRES FORAGE SOAK}$ | 0.115 (0.397) | -0.146, 0.377 | 0.291 (0.032) | 0.026, 0.561 |

**Table 7. Association between experience variables and the likelihood of perceiving weight management as the most important outcome of exercising ponies.**

| Outcome | EXP $_{YEARS}$ | | EXP $_{MET}$ | |
| --- | --- | --- | --- | --- |
| | B (p-value) | 95% CIs | B (p-value) | 95% CIs |
| WMA $_{EXERCISE}$ | -0.297 (0.016) | -0.542, -0.054 | 0.683 (<0.001) | 0.428, 0.945 |

The indirect effect of the relationship between EXP $_{YEARS}$ and WMA $_{EXERCISE}$ was not significant, thus there was no evidence to suggest mediation by CONF $_{DIS}$ (Fig 6).

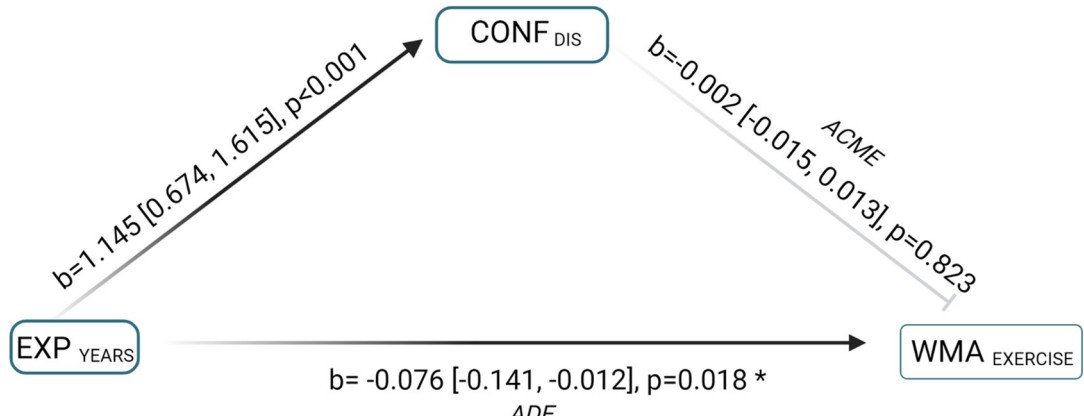

**Fig 6. Mediation model coefficients for the average causal mediation effect (ACME) and average direct effect (ADE) of confidence in ability to identify disease (CONF $_{DIS}$) on the relationship between owners having >20 years' experience caring for native ponies EXP $_{YEARS}$ and their likelihood of perceiving weight management as the most important outcome of exercising their ponies (WMA $_{EXERCISE}$).** CONF $_{DIS}$ did not mediate a significant relationship between EXP $_{YEARS}$ and WMA $_{EXERCISE}$. Figure created with Biorender.com.

and testing preserved forage as mediated by confidence was significant (average direct effect = -0.064, p = 0.039), the average causal mediation effect was not significant (mediation effect = 0.008, p = 0.130).

### 3.4. ≥WMA 4: Exercise

The aspect of exercising ponies that was most frequently selected as being "most important" was the horse's performance, followed by personal enjoyment (Table 1). Eighteen percent of owners felt that weight management was the most important aspect of exercising their ponies. The likelihood of selecting this choice was significantly positively associated with EXP $_{MET}$, while being significantly negatively related to EXP $_{YEARS}$ (Table 7).

## 4. Discussion

Obesity is a substantial welfare issue facing native ponies- with reproductive, orthopaedic and cardiovascular implications [1], as well as being a key risk factor for metabolic diseases. Reducing the prevalence of equine obesity and related conditions requires a better understanding of the human behaviour underpinning horse care. Despite increased attention to the prevalence and risks of obesity, studies of UK horses have shown no drastic decline in its prevalence [3–6, 49]. Previous work has highlighted the need for research to explore causal pathways driving attitudinal and behavioural factors influencing horse owner decision making [32].

The current study found evidence to suggest that having more years of experience in managing native ponies led to increased owner confidence in management and disease identification, but this confidence did not significantly influence the likelihood of those owners' undertaking a selection of weight management approaches (WMAs). Independently of confidence, years of experience had a negative direct effect on owners' likelihood of undertaking WMAs related to forage provision and exercise. In contrast, owners undertaking the nutritional management of metabolic disease related conditions in their ponies were no more or less likely to have greater confidence in either management or disease recognition. However, this group of owners were more likely to undertake positive weight management approaches for their ponies in areas related to seasonal weight management, preserved forage provision, and exercise. These findings are consistent with impressions that management-based

interventions are more likely to be employed by owners after the development of disease -whereas they would be more beneficial to be used in a preventative capacity [50].

In the present study, owners demonstrated high self-reported confidence in their ability to identify obesity in their native ponies, with confidence levels significantly greater in those with ≥ 20 years of experience in native pony management. Confidence in identifying the signs of obesity and laminitis have been equally high in previous reports [50]. Compared with their confidence in identifying colic, loss of condition, laminitis, and obesity, owners tend to have lower self-reported confidence [50], as well as lower self-reported knowledge [51], around equine metabolic syndrome (EMS). Consistent with this, the present work found owners' confidence in identifying laminitis to be comparable with their confidence in identifying obesity, loss of condition and colic, all of which were higher than confidence in identifying metabolic diseases such as EMS and PPID. Together with research which found that that only 1 in 5 surveyed horse owners were familiar with EMS [51], these results could indicate a lack of awareness or understanding amongst owners of the clinical importance, and prevalence, of metabolic disease in native breed type ponies.

The confusion within the horse owning community regarding EMS may stem from its transition from once being primarily associated with obesity to now being recognized as involving ID independent of adiposity [52]. This change in the clinical definition recognizes the potential for insulin dysregulation (ID) and laminitis to occur in non-obese animals [52], although it acknowledges that obesity, when present, can worsen clinical outcomes [15].While separate from EMS, obesity is a modifiable risk factor for ID that owners can learn to identify and monitor. Therefore, risk reduction could be achieved through effective obesity management. Enhancing owner education on equine metabolism and breed-specific management could improve awareness of the heightened metabolic disease risk in obese horses. This then begs the question as to whether improving owner understanding of equine metabolic disease would in turn lead to improved health outcomes for the native ponies under their care. Quantifying owner understanding was outside the scope of the present study, and it cannot be assumed that more years of experience equates to greater knowledge, however it seems plausible that greater understanding would in turn lead to greater confidence. Findings of the present study suggested that confidence did not mediate a relationship between experience and routine use of WMAs, and therefore may not be the appropriate behavioural target for equine obesity related intervention.

The high self-reported confidence in obesity recognition seen presently may have been misplaced, given the high prevalence of obesity in the leisure horse demographic, and known tendency of owners towards weight underestimation in cases of obesity [3, 33, 35, 53]. Instead, a shift toward undertaking some of the WMAs surveyed was more likely where owners had direct experience of managing ponies with metabolic dietary requirements. These findings align with previous work which found that owners engaging in proactive weight management had often experienced a "trigger" event, such as their horse developing laminitis, or a veterinarian pointing out their horses' obesity, prompting them to undertake concerted weight management efforts [34]. Further, research has shown owners to perceive metabolic disease, such as EMS, as having a strong impact upon equine welfare, which may result in a reciprocal negative impact on their own emotional well-being where their own animals have the condition [51]. This evidence seems to support the development of interventions that emphasize the "real-life" impact of obesity in ponies, highlighting the practical and welfare-based benefits (cost, time, improvement in the animals' long-term quality of life) that prevention could afford. Such interventions present an opportunity to highlight the potential adverse effects of obesity, whilst highlighting the value of diagnostic testing for metabolic and endocrine diseases which could help to manage ID and prevent laminitis.

Neither experience, managing a pony with metabolic dietary requirements, nor owner confidence in identifying obesity in their ponies influenced the method or frequency of body condition assessment. Whilst monitoring equine bodyweight and/ or body condition in general was used by the majority of respondents—similar to the proportions found previously [54]—a large proportion of owners used visual, subjective methods. This is in accordance with work showing that owners preferred to adopt visual assessment of their horse's body condition [50]. Owner monitoring of body condition is of vital importance to complement veterinary interventions for equine obesity [1, 2], but continued reliance on non-standardised approaches (e.g. visual assessment) as demonstrated by this study highlights the potential to further promote this aspect of preventative management. The frequent use of standardised methods, such as Henneke's 9—point body condition scoring scale [36], may aid the detection of weight gain and help owners to develop familiarity with the physical assessment of adiposity. Such assessment is particularly important for detecting gradual, seasonal fat accumulation which can increase ponies' risk of laminitis and obesity [25, 55]. However, evidence suggests that some horse owners may find the application of the 9—point body condition scoring system challenging [56]. More objective methods, such as the body condition index, may be more accessible [57], although these methods are not suitable for all native breed type ponies. Our results demonstrated that greater confidence in management and disease recognition was associated with a greater likelihood of undertaking more frequent, standardised methods for monitoring body condition- but this relationship was not driven by accumulated years of pony management, or direct experience of a metabolic disease associated condition. It may be necessary to highlight to owners the need to "update" skills around disease recognition in their ponies to enable them to make timely management adjustments that promote their pony's wellbeing. As changes in body condition occur slowly, the recommended approach would be the combined use of body condition scoring and morphometric measurements, which may give owners greater incentive to consistently monitor their ponies body condition and weight.

The presence of strong seasonal changes in appetite and basal energy requirements within our native breed ponies means they are not suited to modern husbandry methods that can provide year-round access to high quality forages [25]. These seasonal changes in metabolism provide an ideal scenario to support body mass losses during the winter months in the obese animal. There was a higher likelihood of adopting a weight loss approach over the autumn and winter by the owners of ponies with metabolic dietary requirements, indicating an awareness by owners that such management may be beneficial for weight and metabolic disease management. This study also provided evidence of owners using management approaches that could obstruct the natural variations in equine bodyweight [58], where they aimed to maintain their horses' weight across all seasons. Without physical assessment, it is not possible to identify whether owners were maintaining their ponies at a healthy weight. However, extrapolation of previous estimates of obesity in a pony population [6] suggests that in up to 70% of cases, owners could have been aiming to maintain their ponies in overweight or even obese states.

The WMAs relating to preserved forage provision were more likely to be undertaken by owners actively managing ponies with metabolic dietary requirements. Current recommendations suggest that owners should undertake analysis of forage for non-structural and water-soluble carbohydrate content (NSC and WSC respectively) alongside soaking to provide information on the suitability of the forage for the horse [59]. Research has identified a high degree of variability in the results of preserved forage analyses [60], indicating that multiple samples are required from each batch taken for the most representative results. In a survey of 613 UK horse owners who fed haylage, only six out of the twenty-nine who paid for an analysis did so with every batch [61]. This suggests that it may be uncommon for analysis of forage to be undertaken with great frequency, perhaps attributable to high preserved forage bale turnover,

such as on livery yards or where owners feed a large number of horses. With regard to hay soaking, its efficacy in reducing sugar content of preserved forage is dependent upon soaking conditions [62, 63], and owners may be physically unable to adopt this practice due to its physically demanding and time consuming nature [31]. Despite their limitations, hay soaking and forage analysis are measures which could encourage owners to consider the NSC concentrations in preserved forage when making decisions on the appropriate nutrition for their ponies. Whilst not likely to prevent metabolic disease, when used pre-emptively these practices could reduce the risk of exacerbating diagnosed or sub-clinical metabolic disorders through inadvertent provision of high NSC forages [64].

In a recent study involving 504 horse owners in the UK, it was discovered that 11% of participants (n = 57) had conducted an analysis of their horses' preserved forage. This percentage closely aligns with the 14% reported in the present study. Notably, a comparable proportion of respondents in the previous study (16%) and in the current work (20%) were unaware of the practice of forage analysis [61]. It is important to note that the previous research targeted all UK horse and pony owners, whereas the current study specifically to evaluate owners of native-breed type ponies in Scotland. Due to the native breed-type propensity for obesity and ID [5, 10], higher uptake of preserved forage analysis in the present study would have been more promising, and targeted promotion of this practice may be required to reach the owners of native breed types specifically. Agreement between national survey results with those derived from owners in Scotland only, suggests a UK wide need to promote preserved forage analysis as a useful tool to aid the management of ponies at risk of metabolic disease, specifically for monitoring NCS fluctuations in forage sources.

Perception of the value of exercise in the management of metabolic conditions in the horse may have been more positive in owners with experience of the benefits it can confer to horse health. Those managing ponies with EMS, laminitis or overweight were more likely to perceive weight management as the most important outcome of exercise. Combined with dietary restriction, low-intensity exercise can improve insulin sensitivity in obese horses [26], and recent work has highlighted a potentially protective role of exercise against laminitis risk [16]. Given that acutely laminitic animals should not be exercised, the emphasis of this practice in prevention is key. Information regarding ponies' intended purpose was not collected, however, native breed types are typically popular for leisure, breeding and / or competition purposes and our results showed that the horses' performance, and owners' personal enjoyment were the favoured outcomes of exercise. As this was a subjective question without a single preferred answer, the intention was not to suggest that weight management was the most important aspect of exercising ponies. Through asking owners their perspective, the question provided insight on owner priorities in terms of exercising their ponies. The results indicate that the intended purpose of the horse could take precedence over the horses' nutritional and energy-based requirements, leading owners to perceive the weight management aspect of exercise as less important than other areas. Promoting the benefits of incorporating exercise into the management of non-laminitic ponies could help to lower the risk of obesity and metabolic disease, and may encourage a positive change in the perception of exercise as a tool for preventative management.

Ideally, management of native ponies would undergo an overhaul in terms of standard practice, with obesity prevention at the forefront of decision making. Changing human behaviour is the key to achieving animal welfare goals which depend upon human management of nutrition. Within the adapted "Ten Tasks" framework of human behaviour change proposed for meaningful improvements in animal welfare [65, 66], the present work contributes towards understanding the mechanisms relating to horse owner experience and confidence that may help in sustaining current management practices used for native pony management. The

proceeding steps towards reducing obesity in UK equids are to construct behavioural models specific to the equine management, and progress in this area has already been made. Comprehensive behavioural analysis of the human role in equine obesity has shown the complex social environment within the horse management domain [31], and suggests that interventions should be targeted not only toward owner knowledge of obesity, but also toward owner motivation and equestrian social structures. The present findings are in agreement with the concept that generic approaches to improving confidence in weight management are not sufficient. Instead, work towards integrating WMAs, like the 9—point body condition scoring system, forage analysis, exercise, and utilising winter metabolism, into the common vernacular of native pony care could help the shift owner perceptions of obesity.

As for any questionnaire, there is inherent respondent bias that must be acknowledged. In the current study, participants were an experienced group, and a prior interest in laminitis and/or native pony management could have influenced owners' decision to undertake the survey. The present study focused on owners of native ponies in Scotland, and so generalisation of the results to non-native horse care, or to populations outside of Scotland, may be limited. Further, an effect of survival bias may influence the higher likelihood of undertaking WMAs by the owners of native ponies with metabolic disease associated dietary requirements, whose animals may be less likely to have survived had they not altered their management. The mediation analysis adopted in this study is a novel approach to providing key insights into the causal pathways between predictors and outcomes. In order to perform this analysis, variables were dichotomised and grouped in ways which helped to answer the research question at hand, but we acknowledge that such grouping can lead to bias.

Mediation analysis has previously been utilised in the field of canine veterinary medicine as a tool for understanding the role of caregiver burden on the relationship between clinical signs of osteoarthritis, and euthanasia [40]. The application of structural equation modelling associated multivariate techniques to test causal pathways will be an important tool for future research that seeks to understand more about the human factors related to equine disease. Proposed model structures are theoretical but are designed with prior knowledge. There is a risk that theoretical models do not explain the true relationship between the factors assessed, and in testing for causation, there is a risk of masking true effects and relationships which exist outside of our pre-defined structure [67]. Care was taken to consider paths and assess potential confounding relationships, and the limitations of quantitative surveys in analysis of human behaviour are acknowledged. Validated questions relating to measurable behavioural outcomes should be developed to aid future studies in this area.

## 5. Conclusion

The results presented suggest that whilst confidence in care may increase with years of experience, it does not appear to drive the use of management practices that support risk reduction for laminitis, obesity, and EMS. In native pony management, WMAs may be underutilised as a preventative approach, instead being more commonly undertaken by owners after their pony has developed a metabolic health condition of concern. Obesity and laminitis may be perceived by native pony owners as distinct from metabolic disease, demonstrating the need to highlight to owners the interconnected nature of these conditions, and the seriousness of their impact upon pony welfare.

## Supporting information

**S1 File. S1 Table, S1 Fig and Supporting Information S1- additional respondent demographics and management practices information.**
(DOCX)

**S2 File. Survey questionnaire.**
(DOCX)

## Acknowledgments

We wish to gratefully acknowledge the horse owners who took the time to undertake this survey.

## Author Contributions

**Conceptualization:** Patricia A. Harris, Caroline McG. Argo, Christine A. Watson, Madalina Neacsu, Wendy R. Russell, Dai Grove-White, Philippa K. Morrison.

**Data curation:** Ashley B. Ward.

**Formal analysis:** Ashley B. Ward, Patricia A. Harris, Neil M. Burns, Dai Grove-White.

**Funding acquisition:** Caroline McG. Argo.

**Investigation:** Ashley B. Ward, Patricia A. Harris, Caroline McG. Argo, Christine A. Watson, Philippa K. Morrison.

**Methodology:** Ashley B. Ward, Patricia A. Harris, Christine A. Watson, Neil M. Burns, Wendy R. Russell, Dai Grove-White, Philippa K. Morrison.

**Project administration:** Caroline McG. Argo, Madalina Neacsu, Philippa K. Morrison.

**Resources:** Caroline McG. Argo.

**Supervision:** Patricia A. Harris, Caroline McG. Argo, Christine A. Watson, Madalina Neacsu, Wendy R. Russell, Philippa K. Morrison.

**Validation:** Ashley B. Ward.

**Visualization:** Ashley B. Ward.

**Writing – original draft:** Ashley B. Ward, Philippa K. Morrison.

**Writing – review & editing:** Ashley B. Ward, Patricia A. Harris, Caroline McG. Argo, Christine A. Watson, Neil M. Burns, Madalina Neacsu, Wendy R. Russell, Dai Grove-White, Philippa K. Morrison.

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
