## [Decision Letter · Decision Letter 0]

9 Aug 2023

PONE-D-23-11189Confidence does not mediate a relationship between owner experience and likelihood of using weight management approaches for native poniesPLOS ONE

Dear Dr. Ward,

Thank you for submitting your manuscript to PLOS ONE. After careful consideration, we feel that it has merit but does not fully meet PLOS ONE’s publication criteria as it currently stands. Therefore, we invite you to submit a revised version of the manuscript that addresses the points raised during the review process.

Apologies for the slow progress with pushing this manuscript through the review process. It had proven difficult to recruit reviewers. The reviewers comments are favorable and represent minor edits

We look forward to receiving your revised manuscript.

Kind regards,

Chris Rogers

Academic Editor

PLOS ONE

Journal Requirements:

"Co-author PH is employed by the funding organization. This does not alter our adherence to PLOS ONE policies on sharing data and materials. All other authors declare that they have no competing interests. "

5. We note that you have stated that you will provide repository information for your data at acceptance. Should your manuscript be accepted for publication, we will hold it until you provide the relevant accession numbers or DOIs necessary to access your data. If you wish to make changes to your Data Availability statement, please describe these changes in your cover letter and we will update your Data Availability statement to reflect the information you provide

Reviewers' comments:

Reviewer's Responses to Questions

**Comments to the Author**

1. Is the manuscript technically sound, and do the data support the conclusions?

Reviewer #1: Partly

Reviewer #2: Yes

2. Has the statistical analysis been performed appropriately and rigorously? 

Reviewer #1: I Don't Know

Reviewer #2: Yes

3. Have the authors made all data underlying the findings in their manuscript fully available?

Reviewer #1: Yes

Reviewer #2: Yes

4. Is the manuscript presented in an intelligible fashion and written in standard English?

Reviewer #1: Yes

Reviewer #2: Yes

5. Review Comments to the Author

Reviewer #1: This is a well-written manuscript that follows a mostly logical, although overly long in places, progression and addresses an interesting, social-sciences type topic. The social science/behavioural focus of the manuscript differs from the bulk of currently published literature in this research discipline, which increases its novelty. However, I have some concerns that I think should be addressed prior to reconsideration for publication.

Most importantly, the authors seem to conflate obesity with EMS. This is an over-simplification of the two conditions, that while related can also be quite independent from one another. There has been a complete oversight with respect to the critical causative factors for laminitis e.g., hyperinsulinaemia, with repeated suggestion that obesity causes laminitis, which it does not. I think that the authors need to re-focus their manuscript on obesity, and then discuss the related elements, such as laminitis, rather than relying on laminitis to ‘sell’ the gravity of the problem, without adequately explaining the nuances.

The methods and results sections overlap at times, and can be a little confusing as presented. This is not helped by the repetition of some parts of the data between text, tables and figures. Perhaps, more clearly emphasising the novelty of the analysis/approach earlier in the manuscript will help readers less familiar with this type of study to follow your results section.

I have some concerns about the way the respondents were classified, although I understand why it was done this way. Perhaps consider addressing this more robustly, and acknowledging the limitations of your approach. The data is Fig S1 could also help the reader see for themselves the distribution of your responders, so you could consider moving these data to the main manuscript (in turn I would reconsider inclusion of so many of the mediation model coefficient figures as these are not very reader friendly…).

There are some areas of the discussion where I believe that you overinterpret your data, as your discussion points are not supported by P values. Please consider acknowledging the study limitations more carefully throughout.

Lastly, I think that the authors could better characterise their native pony population. This is a very circumscribed cohort of animals living in one part of the world, and surely there are more data to help the reader understand whether these ponies differ from native ponies elsewhere in the UK, and also how they differ from other breeds of horse/pony.

I also have some more minor, specific comments:

Line 61 – the words ‘forms’ seems a bit awkward here. Perhaps types??

Line 98 – needs a full stop

Line 114/5 - this sentence needs a bit of refinement to make better sense wrt to the list items

Line 129 – do you have a reference or some statistical reasoning for 20 years of experience being ‘reasonable’? Does seem a little arbitrary, as someone could have had a hardy native pony at pasture for 20+ years, and not know or care much about it. Whereas an invested owner of 15 years is deemed less experienced?

Line 152 – this is the first mention of PPID. Perhaps worth defining it to some extent in the introduction? Prevalence in the native pony population etc.

Line 170 – check journal requirements for whether you need to define generalised linear model here

Line 172 – a ‘to’ is missing in this sentence

Line 196 – should be ‘a full summary of the sample’ ?

Table 1: is it >20 years or ≥20y? i.e. 20 y or 21y and above? In the table you have grouped to 19 and then 20-29 so do you mean ≥20y? Please clarify

Table 1: would be worth better defining for the reader what the break down is between the variables frequency and method of body condition assessment by the owner. i.e. are most daily assessors using a weight tape or hands or having a quick glance and visual appraisal.

Line 206 – should be ‘than’, not ‘that’

Line 241 on – see me comment above. Seems like some of this is methods and would have been worth explaining before presentation of Table 1. The interaction between type and frequency of monitoring is important to help understand the efficacy of the monitoring approach…

Fig 4: This is some disconnect between the simplicity of your description of the seasonal data and Fig 4. The figure could be re-thought to be clearer for the reader who may have skipped to the figure (and doesn’t read the helpful text)…

Line 370 – I would argue that this point is probably more central to your paper than the general focus of your introduction which is firmly on ID and laminitis. Obesity is not associated with laminitis per se, as obese animals without ID are arguably not at greater risk of laminitis. Perhaps you could re-think some of your introduction to be clearer about the fact that this paper is about assessing attitudes to WMAs and weight loss, and that obesity is the major factor here. While obesity might be a part of EMS, it does not have to be and I think you could make this distinction clearer.

Line 401 – as above, need to parse out the subtleties here, obesity is not a stand alone risk for laminitis

Line 403 – I would say it has firmly evolved…

Line 410 – outside rather than out-with?

Line 443 - a trend is something that occurs over time. I take it you mean you have a P value close to, but over, 0.05?

Line 492 – please reference this statement about usage

Line 504 – I would say that with respect to laminitis prevention veterinary assessment and clinical examination plus laboratory testing for metabolic/endocrine diseases is critically important, far more so than promoting WMAs with no underpinning knowledge of the metabolic health of the individual animal. This is an aspect that you have not addressed (see comments above about conflating obesity and laminitis), and I think it important to clarify what the most important steps should be with respect to laminitis prevntion.

Line 515 – Again I am unsure that obesity directly has any devastating consequences, the same cannot be said for hyperinsulinaemia…..

Reviewer #2: This is an interesting manuscript. I only had a few minor comments which are on the attached edited version of the manuscript. These comments just focus on providing a bit more information on the separation of the owners into two groups and the categorizing associated with body condition score.

6. PLOS authors have the option to publish the peer review history of their article (what does this mean?). If published, this will include your full peer review and any attached files.

Reviewer #1: No

Reviewer #2: No

---

## [Author Response · Author response to Decision Letter 0]

8 Sep 2023

Reviewer #1: This is a well-written manuscript that follows a mostly logical, although overly long in places, progression and addresses an interesting, social-sciences type topic. The social science/behavioural focus of the manuscript differs from the bulk of currently published literature in this research discipline, which increases its novelty. However, I have some concerns that I think should be addressed prior to reconsideration for publication.

Most importantly, the authors seem to conflate obesity with EMS. This is an over-simplification of the two conditions, that while related can also be quite independent from one another. There has been a complete oversight with respect to the critical causative factors for laminitis e.g., hyperinsulinaemia, with repeated suggestion that obesity causes laminitis, which it does not. I think that the authors need to re-focus their manuscript on obesity, and then discuss the related elements, such as laminitis, rather than relying on laminitis to ‘sell’ the gravity of the problem, without adequately explaining the nuances.

The methods and results sections overlap at times, and can be a little confusing as presented. This is not helped by the repetition of some parts of the data between text, tables and figures. Perhaps, more clearly emphasising the novelty of the analysis/approach earlier in the manuscript will help readers less familiar with this type of study to follow your results section.

I have some concerns about the way the respondents were classified, although I understand why it was done this way. Perhaps consider addressing this more robustly, and acknowledging the limitations of your approach. The data is Fig S1 could also help the reader see for themselves the distribution of your responders, so you could consider moving these data to the main manuscript (in turn I would reconsider inclusion of so many of the mediation model coefficient figures as these are not very reader friendly…).

There are some areas of the discussion where I believe that you overinterpret your data, as your discussion points are not supported by P values. Please consider acknowledging the study limitations more carefully throughout.

Lastly, I think that the authors could better characterise their native pony population. This is a very circumscribed cohort of animals living in one part of the world, and surely there are more data to help the reader understand whether these ponies differ from native ponies elsewhere in the UK, and also how they differ from other breeds of horse/pony.

#Author Response 

We are very grateful for your careful consideration of this work and have found your review to be highly instructive for the development of the manuscript. Thank you. 

In regard to the conflation of obesity with EMS, after reviewing your comments we agree that an unintentional suggestion that obesity caused laminitis was present throughout the manuscript. We have applied considerable changes to the introduction and discussion sections in particular to clarify that this is not the case. As recommended, we have placed the central focus on obesity and have provided a fuller explanation of the nuances of the connection between obesity, ID and laminitis, which we hope will adequately address the reviewer’s concerns. 

We have clarified the respondent grouping through including results of an additional sensitivity analysis which provided the justification for dichotomising some groups in the fashion we did. Furthermore, we have added information to the methods section to describe the grouping of approaches to body condition assessment, which we hope will provide a more transparent overview of the data manipulation performed prior to analysis. Figure S1 was transferred to the manuscript (now Fig. 2, line 255), and two of the model diagrams were moved to the supplementary materials (Now Fig S1 and Fig S2). The limitations of grouping variables as performed in this analysis were also highlighted in the discussion section. 

In order to make the distinction between the methods and results section clearer, we have added detail of the novel approach to the analysis in the introduction, and have moved portions of the results into the methods section. 

We have removed sections of the discussion where we agree that links were made which weren’t necessarily supported by the data.

To the reviewer’s final point, we have added a brief explanation of and reference to the genetic and phenotypic differences between native ponies compared to other breeds. 

Line 61 – the words ‘forms’ seems a bit awkward here. Perhaps types??

R- This has been corrected

Line 98 – needs a full stop

R- Full stop added

Line 114/5 - this sentence needs a bit of refinement to make better sense wrt to the list items

R- Restructured sentence

Line 129 – do you have a reference or some statistical reasoning for 20 years of experience being ‘reasonable’? Does seem a little arbitrary, as someone could have had a hardy native pony at pasture for 20+ years, and not know or care much about it. Whereas an invested owner of 15 years is deemed less experienced?

R- Lines 143-151: Thank you for raising this. We have included reference to the sensitivity analysis performed to select the cut off-point which would enable a comparison between more, and less, years of experience. Data from this analysis is now included in Supplementary materials file S1

Line 152 – this is the first mention of PPID. Perhaps worth defining it to some extent in the introduction? Prevalence in the native pony population etc. 

R- Lines 54-56: With respect, we would prefer not to get into any detailed discussion on PPID as this was outside of the scope of the study- but we have introduced it in the introduction

Line 170 – check journal requirements for whether you need to define generalised linear model here

R- Full name added 

Line 172 – a ‘to’ is missing in this sentence

R- Added ‘to’

Line 196 – should be ‘a full summary of the sample’ ?

R- Corrected

Table 1: is it >20 years or ≥20y? i.e. 20 y or 21y and above? In the table you have grouped to 19 and then 20-29 so do you mean ≥20y? Please clarify

R- Thank you for highlighting the discrepancy here. We have clarified this throughout the text. 

Table 1: would be worth better defining for the reader what the break down is between the variables frequency and method of body condition assessment by the owner. i.e. are most daily assessors using a weight tape or hands or having a quick glance and visual appraisal.

R- Lines 281 - 294: Thank you. This summarised description has been added to the text. In addition, Supplementary figure (Figure S1) has been added for clarity. 

Line 206 – should be ‘than’, not ‘that’

R- Corrected

Line 241 on – see me comment above. Seems like some of this is methods and would have been worth explaining before presentation of Table 1. The interaction between type and frequency of monitoring is important to help understand the efficacy of the monitoring approach…

R- Lines 168 - 175: Thank you. Text has been moved to the methods section (lines 168 - 169), and additional information related to the original variables and the approach to dichotomisation has been added (lines 169-175)

Fig 4: This is some disconnect between the simplicity of your description of the seasonal data and Fig 4. The figure could be re-thought to be clearer for the reader who may have skipped to the figure (and doesn’t read the helpful text)…

R- Line 332: Thank you, we agree and have replaced this figure, and the analysis, with a clearer, simple bar chart to summarise these data (line 332). Due to the addition of an earlier figure as recommended, this figure is now Figure 5. 

Line 370 – I would argue that this point is probably more central to your paper than the general focus of your introduction which is firmly on ID and laminitis. Obesity is not associated with laminitis per se, as obese animals without ID are arguably not at greater risk of laminitis. Perhaps you could re-think some of your introduction to be clearer about the fact that this paper is about assessing attitudes to WMAs and weight loss, and that obesity is the major factor here. While obesity might be a part of EMS, it does not have to be and I think you could make this distinction clearer.

R- Lines 50-61: Thank you for highlighting this. We have altered paragraph 1 introduction to focus more on the direct impacts of obesity in address to your points. 

Line 401 – as above, need to parse out the subtleties here, obesity is not a stand alone risk for laminitis

R- Corrected 

Line 403 – I would say it has firmly evolved…

R- Corrected 

Line 410 – outside rather than out-with?

R- Corrected 

Line 443 - a trend is something that occurs over time. I take it you mean you have a P value close to, but over, 0.05?

R- Corrected 

Line 492 – please reference this statement about usage

R- Lines 500 - 504: We have added a description of forage usage and cited a study which was published after the initial drafting of this manuscript. We have rephrased, and supported the statement with the reference: 

61. Moore-Colyer M, Westacott A, Rousson L, Harris P, Daniels S. Where Are We Now? Feeds, Feeding Systems and Current Knowledge of UK Horse Owners When Feeding Haylage to Their Horses. Animals. 2023;13: 1280. doi:10.3390/ani13081280

Line 504 – I would say that with respect to laminitis prevention veterinary assessment and clinical examination plus laboratory testing for metabolic/endocrine diseases is critically important, far more so than promoting WMAs with no underpinning knowledge of the metabolic health of the individual animal. This is an aspect that you have not addressed (see comments above about conflating obesity and laminitis), and I think it important to clarify what the most important steps should be with respect to laminitis prevntion.

R- Lines 453 - 455: Thank you, this has been corrected and we agree that the importance of veterinary diagnosis of metabolic and endocrine disease was not adequately acknowledged previously. We have added text to this point. 

Line 515 – Again I am unsure that obesity directly has any devastating consequences, the same cannot be said for hyperinsulinaemia…..

R- Thank you for highlighting this. We have adapted the discussion throughout to avoid overstating the impact of obesity specifically in relation to metabolic disease

Reviewer #2: This is an interesting manuscript. I only had a few minor comments which are on the attached edited version of the manuscript. These comments just focus on providing a bit more information on the separation of the owners into two groups and the categorizing associated with body condition score

R- We thank you for your time taken to consider and comment upon this manuscript, and for your interest in this work. In address to the comments relating to categorising owners into groups based on years of experience and approach to body condition score, we have added details of a sensitivity analysis to support the grouping of respondents by years of experience. Data from this analysis is now included in Supplementary materials file S1, with a summary of reasoning in lines 143-151.

Further, in relation to the categories of body condition scoring assessment , we have added details and a fuller description of the grouping of body condition scoring methods (lines 168 - 175). A supplementary figure (Figure S1) has also been added to aid interpretation of these data. 

We agree that these details will greatly improve the clarity of the manuscript and we thank you for highlighting these points.

---

## [Editor Report · Decision Letter 1]

2 Oct 2023

Confidence does not mediate a relationship between owner experience and likelihood of using weight management approaches for native ponies

PONE-D-23-11189R1

Dear Dr. Ward,

We’re pleased to inform you that your manuscript has been judged scientifically suitable for publication and will be formally accepted for publication once it meets all outstanding technical requirements.

Kind regards,

Chris Rogers

Academic Editor

PLOS ONE

Additional Editor Comments (optional):

Thank you for the edits to the manuscript in response to the reviewers comments. This is an interesting manuscript and it may now proceed with the publication process.
---

## [Editor Report · Acceptance letter]

5 Oct 2023

PONE-D-23-11189R1 

Confidence does not mediate a relationship between owner experience and likelihood of using weight management approaches for native ponies 

Dear Dr. Ward:

I'm pleased to inform you that your manuscript has been deemed suitable for publication in PLOS ONE. Congratulations! Your manuscript is now with our production department. 

Kind regards, 

on behalf of

Dr. Chris Rogers 

Academic Editor

PLOS ONE